# Cognitive Flexibility in Mice: Effects of Puberty and Role of NMDA Receptor Subunits

**DOI:** 10.3390/cells12091212

**Published:** 2023-04-22

**Authors:** Lisa Seifried, Elaheh Soleimanpour, Daniela C. Dieterich, Markus Fendt

**Affiliations:** 1Institute for Pharmacology and Toxicology, Otto-von-Guericke University Magdeburg, D-39120 Magdeburg, Germany; 2Center of Behavioral Brain Sciences, Otto-von-Guericke University Magdeburg, D-39120 Magdeburg, Germany

**Keywords:** adolescence, attentional set shifting task, cognitive flexibility, NMDA receptor subunits, orbitofrontal cortex, puberty

## Abstract

Cognitive flexibility refers to the ability to adapt flexibly to changing circumstances. In laboratory mice, we investigated whether cognitive flexibility is higher in pubertal mice than in adult mice, and whether this difference is related to the expression of distinct NMDA receptor subunits. Using the attentional set shifting task as a measure of cognitive flexibility, we found that cognitive flexibility was increased during puberty. This difference was more pronounced in female pubertal mice. Further, the GluN2A subunit of the NMDA receptor was more expressed during puberty than after puberty. Pharmacological blockade of GluN2A reduced the cognitive flexibility of pubertal mice to adult levels. In adult mice, the expression of GluN2A, GluN2B, and GluN2C in the orbitofrontal cortex correlated positively with performance in the attentional set shifting task, whereas in pubertal mice this was only the case for GluN2C. In conclusion, the present study confirms the observation in humans that cognitive flexibility is higher during puberty than in adulthood. Future studies should investigate whether NMDA receptor subunit-specific agonists are able to rescue deficient cognitive flexibility, and whether they have the potential to be used in human diseases with deficits in cognitive flexibility.

## 1. Introduction

Cognitive flexibility is defined as the ability to adapt behavior to changes in the environment [1]. It is considered to be one of the processes of executive functions and is mediated by different subregions of the frontal cortex [2]. Different neuropsychiatric and neurological disorders, such as schizophrenia, attention deficit hyperactivity disorder, or Alzheimer’s disease, but also healthy aging, are associated with impaired cognitive flexibility [3,4,5].

In laboratory rodents, cognitive flexibility can be measured by the attentional set shifting task (ASST) [6,7]. The ASST is based on discrimination learning, first simple and later compound discrimination, in which a second stimulus dimension must be ignored. This is followed by reversal phases, in which the contingencies of the reward-associated stimuli are changed, as well as intra- and extra-dimensional shifts, in which stimulus exemplars are changed and the new reward-indicating stimulus comes either from the same stimulus dimension as before or from the previously ignored stimulus dimension, respectively [8].

Studies in humans and laboratory rodents have shown that cognitive flexibility is sensitive to pharmacological interventions targeting the NMDA receptor. NMDA receptor antagonists such as phencyclidine, dizocilpine, or ketamine impair cognitive flexibility, e.g., as measured by the ASST [9,10], whereas NMDA receptor agonists such as D-cycloserine improve cognitive flexibility [11]. These effects are most likely mediated by NMDA receptors in the frontal cortex, as altered NMDA receptor signaling in the frontal cortex has been found in conditions of impaired cognitive flexibility [12,13]. Of note, NMDA receptor signaling is determined in part by the composition of its subunits [14]. Indeed, NDMA receptors are tetramers composed of two GluN1 and two GluN2 (or GluN3) subunits. A reduction or loss of GluN2A or GluN2B, for example, leads to deficits in cognitive flexibility [15,16]. In schizophrenia patients, reduced expression of GluN2A, hypofunction of the NMDA receptor, and impaired cognitive flexibility are found [3,17]. While these data emphasize an important role of particular NMDA receptor subunits in conditions of impaired cognitive flexibility, less is known whether NMDA receptor subunits also play a role in conditions of high cognitive flexibility.

In humans, high cognitive flexibility is observed during puberty [18,19]. In laboratory rodents, the findings are inconsistent [20], with some studies reporting higher cognitive flexibility during puberty or adolescence [21,22], while others find lower cognitive flexibility [23,24,25]. Interestingly, puberty is also the phase when the expression of different NMDA receptor subunits such as GluN2A, GluN2B, and GluN2C reaches a plateau or even a peak [26]. Whether and how NMDA receptor subunit expression is related to altered cognitive flexibility during puberty is not yet known.

The present study examined whether cognitive flexibility changes in female and male mice during puberty compared with adulthood and whether such changes are related to altered NMDA receptor subunit expression (GluN2A, GluN2B, GluN2C) in the orbitofrontal cortex (OFC), a region crucial for cognitive flexibility [13,16,27]. Subsequently, such a potential relation should be tested by pharmacological intervention. The present findings show increased cognitive flexibility, as measured by the ASST, in adolescent mice. This effect was more pronounced in female mice. In naive mice, expression of GluN2A but not GluN2B and GluN2C was increased in the OFC during puberty. Treatment with the GluN2A-specific antagonist PEAQX impaired ASST performance in pubertal mice. Together, these data indicate that GluN2A expression in the OFC is associated with higher cognitive flexibility during puberty.

## 2. Materials and Methods

### 2.1. Animals

For the experiments, we used female and male C57BL/6J mice from the institute’s own breeding colony (original breeding stock from Charles River, Sulzfeld, Germany). Mice were housed in unisex groups under controlled conditions of humidity (50–55%), temperature (22 ± 2 °C) and 12:12 light/dark cycle (lights-on: 6:00–18:00). All experiments were conducted during the lights-on period. Tap water and normal rodent chow were provided ad libitum; however, before and during the attentional set shifting task, food restriction was performed (see below). The experiments were performed according international guidelines of animal care and use for experimental procedures (2010/63/EU) with confirmed ethical approval (Landesverwaltungsamt Sachsen-Anhalt, Az.42502-2-1618 Uni MD).

### 2.2. Substance

PEAQX (NVP-AAM077) tetrasodium hydrate (MedChemExpress via Hölzel Diagnostika GmbH, Cologne, Germany), an antagonist of the GluN2A subunit of N-methyl-D-aspartate (NMDA) receptors [28], was dissolved in saline. Mice received i.p. injections of either 32 mg PEAQX/kg body weight or saline (injection volume: of 10 µL/g body weight) 20 min before the experiment started. The used dose, injection volume, administration route and time was based on literature [29,30,31].

### 2.3. Attentional Set Shifting Task (ASST)

In the ASST, mice were trained to differentiate a pair of relevant cues, one of them predicting a reward in a small bowl, and to ignore a pair of irrelevant cues. These cues were either visual/haptic (filling medium of the bowl) or olfactory (odorants, put on a filter paper at the outside of the bowl). 

#### 2.3.1. Setup and Material

The ASST box (41 cm × 22 cm × 24 cm; custom-made, University of Magdeburg, Germany) consisted of a waiting compartment and two choice compartments separated from each other by a transparent wall. Between the waiting compartment and the choice compartments, there were transparent sliding doors that could be opened by the experimenter. In each of the compartments, a bowl (5 cm diameter, 2.5 cm height) was placed. In the waiting compartment, one bowl was filled with water, while in the choice compartments, the bowls were filled with medium and the reward (chocolate rice, ca. 20 mg, Nordgetreide GmbH & Co. KG, Lübeck, Germany).

As filling media six different exemplars were used: M1: big green beads, M2: small green beads, M3: big grains (grains = deco gravel), M4: small grains, M5: big clay granulate, M6: small clay granulate. The exemplars of the odors were 30 µL of different odorants (1:20 dissolved in paraffin oil): O1: citral, O2: eucalyptol, O3: R-(+)-carvone, O4: valeric acid, O5: R-(-)-carvone, O6: 2-phenylethanol (all odorants were purchased from Sigma-Aldrich, Darmstadt, Germany).

#### 2.3.2. Food Restriction

On day one of the experiment, mice were handled, their basal bodyweight was recorded and moderate food restriction began until the end of the experiment (see Figure 1). Adult mice were food restricted to maintain 90–95% of their free feeding body weight. Pubertal mice underwent very moderate food restriction with the goal of maintaining their body weight. The amount of food was based on the actual individual body weight of the mice and was given to the mice every day after the behavioral experiment.

#### 2.3.3. Habituation to the Bowls and the Reward

On the second day, the habituation to the bowls and reward started. Two reward-containing bowls were placed in their home cage. Choco rice was distributed in the cage, but also at the top and bottom of the bowls, which were filled with bedding material. Usually, mice learn within hours to eat the reward and to dig for it in the testing bowls.

After a break of 2 days, we went on with the habituation in the testing box. 

#### 2.3.4. Group Habituation

On the third experimental day, all mice of one cage were put into the ASST box (containing three bowls, two of them with reward, one of them with water) and had 30–45 min time to explore the box and learn to dig in the bowls. 

#### 2.3.5. Single Habituation

One hour later, the mice were put alone into the ASST box. This time, only one of the two testing bowls contained the reward. In the first four trials, the reward was placed on top of the bedding material in the bowl; in the next four trials the reward was placed gradually deeper in the bowl until the reward was placed on the bottom of the testing bowl in the last four trials. After successful trials (finding and eating a reward), the reward was replaced. Usually, mice learned with 30–60 min to find the rewards. Of note, the unbaited bowl was sprinkled with Choco Rice powder to prevent the mice from using the smell to retrieve the reward. 

#### 2.3.6. Test Days

The actual ASST is composed of 7 phases performed on 4 consecutive days:Day 1, simple discrimination (SD): In this phase, only cues of one dimension were presented, either medium or odor. The goal of this phase was that mice learn which of the two media or odors, respectively, is associated with the reward. For example, big green beads (M1) indicated the reward while small green beads (M2) were not associated with reward.Day 2, compound discrimination (CD) and reversal (Rev1). Importantly, cues of the second dimension were added. However, still the cues of the SD phase indicate the reward, i.e., the cues of the second dimension were irrelevant. For example, the big green beads (M1) were still rewarded, but were either presented with citral (O1) or eucalyptol (O2). In the Rev1 phase, the previously not rewarded cue was now the rewarding one (in our example the small green beads (M2)).Day 3, intra-dimensional shift (IDS) and reversal (Rev2). All cue exemplars were exchanged but the relevant cue dimension stayed the same. For example, big grains (M3) were rewarded, while small grains (M4) were not. Irrelevant since not associated with the reward were the new odorants valeric acid (O3) and R-(+)-carvone (O4). In Rev2, contingencies changed and the small grains (M4) were rewarded.Day 4, extra-dimensional shift (EDS) and reversal (Rev3). Again, new cue exemplars were used. However, now the relevant cue dimension changed, i.e., the odorants became relevant. For example, R-(-)-carvone (O5) predicted the reward, 2-phenylethanol (O6) not. The new filling media big clay granulate (M5) and small clay granulate (M6) were irrelevant. In Rev3, the contingency of the two odors changed, so 2-phenylethanol (O6) was rewarded.

At the beginning of each trial, the mouse was in the waiting compartment. Then, the choice compartments were equipped with the bowls that were previously prepared outside the box. The side of the rewarded bowl as well as the combinations of filling media and odorants changed from trial to trial in a pre-defined pseudo-randomized order. When the sliding doors opened, the mouse had the opportunity to explore both choice compartments with the two bowls. When the mouse started to dig, the sliding doors were closed and the choice of the mouse was recorded.

Of note, there were four “free trials” at the beginning of the SD, CD and IDS phases, in which the mice could correct their decision, i.e., leaving the wrong choice compartment after starting to dig. This was to make sure that the mouse had enough time exploring the cue exemplars. We considered a phase as successfully completed if the mouse made six consecutive correct trials. For each of the phase, the number of trials and the number of errors until the mouse reached the criteria of six consecutive correct choices was noted. Within and between the different groups (age, sex, treatment), the starting cue dimensions (medium, odor) as well as the starting cue was pseudo-randomized.

### 2.4. Collecting Brain Samples and Molecular Analyses

#### 2.4.1. Brain Sample Preparation

Mice were anesthetized with isoflurane (CP-Pharma, Burgdorf, Germany) and decapitated. In the case of mice with previous behavioral experiments, this was done immediately after completing the last phase of the ASST. Whole brains were quickly extracted and coronal brain dissection (1 mm thick) was performed using the mouse coronal brain slicer matrix (RBM-2000C; ASI-instrument, Warren, MI, USA). The brains were dissected on ice into different regions; for the present study, the OFC was used. Samples were stored at −80 °C until used for Western blot analysis.

#### 2.4.2. Crude Synaptosomal Membrane Preparation

The crude synaptosomal fraction was extracted from the orbitofrontal cortex. Samples were homogenized in ice-cold 0.32 M sucrose solution containing 5mM HEPES pH 7.4, protease inhibitors (cOmplete™ Protease Inhibitor Cocktail, Roche #04693132001; Merck, Darmstadt, Germany), and phosphatase inhibitors (PhosSTOP™, Roche #04906837001; Merck, Darmstadt, Germany). The homogenate was prepared using glass potter S (Sartorius, Göttingen, Germnay) with a Teflon pestle (B. Braun AG, Melsungen, Germany) at 1000 rpm. The homogenate was centrifuged at 1000× *g* for 5 min at 4 °C, the P1 pellet consisting of nuclei was discarded and the supernatant (S1) was further centrifuged at 12,000× *g* for 20 min at 4 °C. The supernatant (S2) was discarded and the pellet (p2) was resuspended in sodium dodecyl sulfate (SDS). Protein quantification was measured using amido black protein assay.

#### 2.4.3. Western Blot Analysis

Automated Western blot analysis was performed using Jess™ Simple Western (Protein simple; Biotechne, Wiesbaden, Germany). Samples were diluted to 0.2 mg/mL using 0.1x sample buffer (Protein simple, #042-195; Biotechne, Wiesbaden, Germany) and 5 μL were loaded to each well in the 12-230kDa Jess separation module (protein simple, #SMW004; Biotechne, Wiesbaden, Germany). Antibodies used in this study were as follows: Anti-NMDAR2A (1:50, #AB1555P; Millipore, Darmstadt, Germany), anti-NMDAR2B (1:50, #ab93610; Abcam, Berlin, Germany), anti-NMDAR2C (1:25, #NB300-107; Novus, Gudensberg, Germany). Secondary anti-rabbit (Protein simple, #DM-001; Biotechne, Wiesbaden, Germany) and secondary anti-mouse (Protein simple, #DM-002; Biotechne, Wiesbaden, Germany) antibodies in addition to luminol-peroxidase mix were used according to the manufacturer’s instruction. The expression level of each protein was normalized to the total protein using the Replex™ module (Protein simple, #RP-001) and the total protein detection module (Protein simple #DM-TP01; Biotechne, Wiesbaden, Germany). Data analysis was performed with Compass SW software (Protein simple; Biotechne, Wiesbaden, Germany).

### 2.5. Experiments

#### 2.5.1. Experiment 1: ASST Performance during Puberty and Adultness

Pubertal and adult mice were tested in ASST according to the protocol described above. Pubertal mice started the ASST on postnatal day 28 (PD28), the adult mice at an age of 8 weeks (Figure 1).

#### 2.5.2. Experiment 2: NMDA Receptor Subunits Expression

To assess the expression of the GluN2A, GluN2B and GluN2C subunits of the NMDAR, brains of naive mice were collected on PD21, PD29 and PD43-45 (before, during, and after puberty; Figure 1). In addition, the brains of the mice in experiment 1 were collected immediately after the last ASST phase. Then, the brain samples were analyzed as described above.

#### 2.5.3. Experiment 3: GluN2A Blockade during ASST in Pubertal Mice

Based on the findings of experiment 1 and 2, we pharmacologically blocked the GluN2A subunit of the receptor to characterize its importance for the ASST-performance. Thus, only pubertal mice were tested, with half of them having a receptor blockade and the other half not. The mice received an injection on each day of the ASST, 20 min before the start of the experimental session (Figure 1).

### 2.6. Statistical Analyses

In addition to the number of trials and the number of errors until the criterion of six consecutive correct trials, we also classified the errors in the reversal phases into perseverative and regressive types. Using logistic regression, we calculated the number of trials after which the mice made correct decisions with a probability greater than 50%. Errors before this trial were defined as perseverative, i.e., the mouse still followed the old rule. Errors after this trial were defined as regressive, i.e., the mouse was already in the process of acquiring the new rule [32].

Data analysis was performed using SYSTAT 13 (Systat Software GmbH, Düsseldorf, Germany.) and Prism 8.0 (GraphPad Software Inc., La Jolla, CA, USA). Analysis of variance (ANOVA) was performed, followed by post hoc multiple comparisons using the two-stage linear step-up procedure of Benjamini, Kriger and Yekutieli.

## 3. Results

### 3.1. Experiment 1: ASST Performance during Puberty and Adultness

Behavioral performances in the ASST, i.e., number of trials to the criterion of six consecutive correct trials (Figure 2a–c), were analyzed with a multi-factorial ANOVA using sex and age of the mice as between-subject factors and phase of the ASST as within-subject factors. As expected, the performance of the mice differed in the different ASST phases (factor phase: *F*_6,126_ = 13.05, *p* < 0.001). The Rev1 and Rev2 was generally more difficult than the phases before (post hoc comparisons: ps < 0.04), and the EDS was more difficult than the IDS (p = 0.45). Of note, the factor phase did not interact with the other factors (*F*s < 0.91, *p*s > 0.45). Importantly, ASST performance was improved in pubertal mice (factor age: *F*_1,21_ = 27.86, *p* < 0.001; Figure 2a). This effect was more pronounced in the female mice than in male mice (interaction age x sex: *F*_1,21_ = 4.94, *p* = 0.037), however, sex had no main effects (*F*_1,21_ = 0.09, *p* = 0.77). Post hoc comparisons showed that puberty improved all phases in female mice except the SD phase (*p*s < 0.05; Figure 2b) while in male mice only the IDS phase was significantly improved (*p* = 0.02; Figure 2c).

The analysis of the errors to criterion confirmed the beneficial effects of puberty (*F*_1,21_ = 17.36, *p* < 0.001; Figure 2d–f). Sex did not interact with age (*F*_1,21_ = 2.84, *p* = 0.11) and had no main effects (*F*_1,21_ = 0.11, *p* = 0.74; Figure 2d). However, post hoc comparisons revealed that the effects of puberty were more pronounce in females (*p* < 0.05 in Rev1, IDS, EDS, Rev3; Figure 2e) than in males (*p* = 0.01 in IDS; Figure 2f). For the reversal phases, we also analyzed the number of perseverative and regressive errors (Figure 2g–j). During puberty, there was only a trend for a reduction in perseverative errors (*F*_1,21_ = 2.97, *p* = 0.099; Figure 2g,h), while regressive errors were significantly decreased (*F*_1,21_ = 9.46, *p* = 0.006; Figure 2i,j). Again, the effects of puberty were more pronounced in female mice (*p*s < 0.04 in Rev 2, Rev 3).

### 3.2. Experiment 2: NMDA Receptor Subunits Expression before, during and after Puberty

Figure 3 depicts the expression of the NMDA receptor subunits GluN2A, GluN2B, and GluN2C (horizontal panels) in the OFC from experimentally naive mice before, during, and after puberty (first vertical panel), and from ASST-experienced mice, i.e., the mice of experiment 1 (second vertical panel). Further, the association of the expression levels with the overall ASST performance is shown for pubertal and adult mice (third or fourth vertical panel, respectively). For the sake of clarity and since we did not find any interactions of the factor sex with the other factors, we pooled the data, but indicated the sex of the mouse for each individual measure.

The expression of GluN2A in experimentally naive mice was affected by the factor age (*F*_2,19_ = 4.42, *p* = 0.03; Figure 3a). In detail, expression in adult mice was significantly lower than during puberty (*t* = 2.91, *p* = 0.03). The sex of the mice had no effects and did not interact with age (*F*s < 3.20, *p*s > 0.09). In ASST-experienced mice, the effect of age was opposite, i.e., adult mice expressed more GluN2A than pubertal mice (*F*_1,16_ = 5.58, *p* = 0.03; Figure 3b). Last, linear regression analysis using the individual mean ASST performance (trials to criterion) and the individual GluN2A expression in the OFC was performed. There was no correlation in pubertal mice (*r*^2^ = 0.02, *p* = 0.63; Figure 3c) but a negative correlation in adult mice (*r*^2^ = 0.59, *p* = 0.03; Figure 3d). The more GluN2A in adult mice were expressed, the fewer trials to criterion were needed, i.e., the better ASST performance.

The effect of age on GluN2B expression was different. In naive mice, expression was also affected by age (*F*_2,21_ = 5.84, *p* = 0.01; Figure 3e). GluN2B expression was highest before puberty and then decreased. After puberty, there was significantly lower GluN2B expression than before puberty (*t* = 4.82, *p* = 0.007). Generally, there was higher GluN2B expression in male mice (*F*_1,21_ = 5.59, *p* = 0.03) but sex did not interact with age (*F*_1,21_ = 1.33, *p* = 0.29). In ASST-experienced mice, expression did not differ between pubertal and adult mice (*F*_1,16_ = 2.61, *p* = 0.13; Figure 3f) but expression was higher in female than in male mice (*F*_1,21_ = 6.99, *p* = 0.02; interaction: *F*_1,21_ = 2.75, *p* = 0.12). In pubertal mice, the individual GluN2B expression was not correlated with the ASST performance (*r*^2^ = 0.01, *p* = 0.67; Figure 3g) but in adult mice, increased GluN2B expression was associated with fewer trials to criterion, i.e., better ASST performance (*r*^2^ = 0.76, *p* = 0.005; Figure 3h).

GluN2C expression was not affected by age, neither in naive mice (*F*_2,21_ = 0.16, *p* = 0.85; Figure 3i), nor in ASST-experienced mice (*F*_1,21_ = 0.02, *p* = 0.89; Figure 3j). There was no effect of sex in naive mice (*F*_1,21_ = 0.57, *p* = 0.57). However, female ASST-experience mice had much higher GluN2C expression than male mice (*F*_1,13_ = 53.73, *p* > 0.0001). In both pubertal and adult mice, higher GluN2C expression was associated with fewer trials to reach the criterion, i.e., increased performance (pubertal mice: *r*^2^ = 0.616, *p* = 0.004; Figure 3k; adult mice: *r*^2^ = 0.67, *p* = 0.048; Figure 3l).

### 3.3. Experiment 3: GluN2A Blockade during ASST in Pubertal Mice

As in the previous behavioral experiment, an ANOVA was conducted with sex and treatment as between-subject factors and ASST phase as within-subject factor. Analysis of the trials to criterion (Figure 4a–c) revealed an effect of ASST phase (*F*_6,96_ = 48.65, *p* < 0.0001) and no interactions of phase with the other factors (*F*s < 0.88, *p*s > 0.18). Performance in Rev1 and Rev2 was significantly worse than in the previous phases and EDS was more difficult than IDS (*p*s < 0.001). Treatment with the GluN2A-specific antagonist PEAQX significantly increased the number of trials (*F*_1,16_ = 104.40, *p* < 0.001). This effect was not specific to an ASST phase (interaction treatment x phase: *F*_6,96_ = 0.41, *p* = 0.87) or sex (interaction treatment x sex: *F*_1,16_ = 0.07, *p* = 0.79). The latter was confirmed by post hoc comparisons, which revealed significantly more trials to criterion after PEAQX treatment in five phases of ASST in both sexes (females: p < 0.02 for SD, CD, Rev1, IDS, Rev2; males: *p* < 0.05 for CD, Rev1, IDS, Rev2, EDS). In addition, there was a main effect of sex (*F*_1,16_ = 11.26, *p* = 0.004), i.e., female mice (Figure 4b) generally required less trials to criterion than male mice (Figure 4c).

Analysis of the error to criterion (Figure 4d–f) fully confirmed these effects of PEAQX. There was an increase in errors after PEAQX treatment (*F*_1,16_ = 79.32, *p* < 0.001) that was neither phase (interaction treatment x phase: *F*_6,96_ = 1.28, *p* = 0.27) nor sex specific (interaction treatment x sex: *F*_1,16_ = 0.06, *p* = 0.81). In addition, PEAQX increased both perseverative (*F*_1,16_ = 30.24, *p* < 0.001; Figure 4g,h) and regressive errors (*F*_1,16_ = 35.20, *p* < 0.001; Figure 4i,j).

## 4. Discussion

The importance of NMDA receptors for cognitive processes is undisputed. Disease- or age-related NMDA receptor dysfunction or downregulation, as well as pharmacological antagonism are associated with or lead to cognitive deficits [33,34]. This makes the NMDA receptor an attractive target for drugs designed to rescue impaired cognitive function [35]. However, the biology of the NMDA receptor is complex and far from fully understood [36]. This also includes its role in cognitive flexibility. There is no doubt that impaired cognitive flexibility is related to NMDA receptor dysfunction [12,13]. The usual approach to investigating the role of a receptor in cognition is to examine its role under conditions of cognitive deficits. However, the present study took the approach to focus on a condition of high cognitive flexibility, i.e., puberty [18,19,20,21,22], and investigated the role of the different NMDA receptor subunits in high cognitive flexibility during this developmental stage.

In our first experiment, we submitted pubertal and adult mice to the ASST. The ASST was prepared by a phase of food restriction, habituation, and pre-training, lasting together four days. After a weekend break, the four remaining days of the ASST followed. For the pubertal mice, habituation started immediately on the day after weaning (PD21), i.e., the ASST was performed from PD28 to PD31. This means that the ASST was performed for both female and male mice during the phase of puberty, which is defined by the rise in sex hormones at the onset and sexual maturity at the completion [20,37]. The adult mice in our study were approximately 8 weeks (56 days) old at the start of the ASST, which is well after the offset of puberty (PD35-PD40) and adolescence (PD40-50 [37]). We found that ASST performance of pubertal mice was significantly better than that of adult mice (Figure 2). This effect was more pronounced in female mice than in male mice. Of note, we analyzed ASST performance not only by the number of trials and the number of errors until the criterion of six consecutive correct trials was reached, but also categorized the types of error made by the mice. Logistic regression analyses were used to calculate the trial number above which the probability of a correct decision was higher than 50%. Errors made before this trial were defined as perseverative, while errors made after this trial were defined as regressive [32]. While age had no effects on the number of perseverative errors, the number of regressive errors was increased in adults compared with pubertal mice, mainly driven by the female mice. This increase in regressive errors in adult mice resulted in a balanced number of perseverative and regressive errors, whereas pubertal mice made approximately twice as many perseverative as regressive errors. Such a bias to perseverative errors were already reported for laboratory rodents with cocaine or ethanol exposure during adolescence [38,39] but also generally for female mice [40].

Our findings that cognitive flexibility is higher during puberty than during adulthood are only partially consistent with published data. Two of the published studies investigating cognitive flexibility in pubertal, juvenile or adolescent, respectively, mice used only male mice [21,24] whereas one study investigated both sexes [23]. The latter study found decreased cognitive flexibility in male adolescent mice compared with adult mice but no differences in females, while the two studies with male mice found both increased [21] and decreased [24] cognitive flexibility in juvenile or adolescent mice. In all three studies, the mice were approximately one week older than the mice of the present study, which is very likely after puberty. This small age difference may be crucial, as human studies have shown that cognitive flexibility gradually declines with age [18] and the present study shows a decrease in GluN2A levels after puberty (Figure 3). However, as mentioned in the introduction, highest cognitive flexibility in humans is found during puberty (12-14 years) as well as during early childhood (4 years) [18,19]. Regarding sex differences, there are very few human studies that have examined this topic, and they found no difference or rather lower cognitive flexibility in female pubertal participants compared to male subjects [18,41]. Similar data have been published from laboratory rodent, where females, for example, are more sensitive to the impairing effects of stress on cognitive flexibility [42]. Taken together, the present study demonstrated, in contrast to some published findings, higher cognitive flexibility in pubertal mice compared with adult mice. This effect of puberty was more pronounced in female mice.

The aim of the second experiment was to evaluate whether the increased cognitive flexibility during puberty was associated with changes in the expression of distinct NMDA receptor subunits (GluN2A, GluN2B, GluN2C) in the orbitofrontal cortex (OFC). The OFC is a frontal cortex region crucial for cognitive flexibility, especially reversal learning [13,16,27]. We decided to focus our molecular analyses on the OFC since puberty had very pronounced effects in reversal learning. To analyze the expression of NMDA receptor subunits, we used Jess™ Simple Western, i.e., automated capillary-based protein separation and immunodetection, with a very high specificity [43]. The first analyses were performed in naive mice, before, during, and after puberty. We found that GluN2A expression peaks during puberty and that there is significantly different expression in the OFC between puberty and adultness (Figure 2). For GluN2B, we observed a decline in expression from before via during until after puberty. However, there was no difference in its expression between during and after puberty. Further, there were no obvious changes in GluN2C. Next, we analyzed the expression of NMDA receptor subunits in the mice of experiment 1, i.e., pubertal and adult mice that were submitted to ASST. For GluN2A, we measured lower expression in ASST-experienced pubertal mice than in adult mice, whereas GluN2B and GluN2C expression did not differ.

When comparing the expression levels in naive and ASST-experienced pubertal mice, it appears at first glance that the ASST experience affected the expression of NMDA receptor subunits. GluN2A seems to be down-regulated in ASST-experienced mice, while GluN2C seems to be upregulated in ASST-experienced mice. However, the study design does not allow such comparisons since naive mice were really experimentally naive, whereas ASST-experienced mice did not only had the ASST experience with several cognitive challenges, but were also handled, had several habituation procedures (setup, reward), were food-deprived, and got tasty rewards. Both, the cognitive challenges but also the other mentioned procedures can affect the expression of NMDA receptor subunits [44,45,46,47].

To better understand how ASST performance and NMDA receptor subunit expression are associated, we further investigated whether the expression of the different NMDA receptor subunits is correlated with ASST performance. For GluN2A and GluN2B, we found no correlation in pubertal mice, but higher expression of both subunits in the OFC of adult mice was positively correlated with better ASST performance. For GluN2C, such a correlation was found in both pubertal and adult mice. Together, these findings indicate that, at least in adult mice, a high expression of GluN2A, GluN2B, and GluN2C in the PFC is beneficial for cognitive flexibility. This is in line with previously published studies that found, for example, a correlation between GluN2B expression in the frontal cortex and spatial learning [48]. Interestingly, also reversal learning was investigated in the latter study, but it was not analyzed whether there was an association between expression levels and reversal learning performance. Of note, we did not see a correlation of expression levels of all NMDA receptor subunits with ASST performance in pubertal mice. Only for GluN2C, there was such a positive correlation, but this could be caused by sex-dependent effects of ASST experience on GluN2C expression.

Based on the finding that pubertal and adult mice only differ in GluN2A expression, we performed a third experiment, in which we treated pubertal mice with the GluN2A-specific antagonist PEAQX. Treatment with PEAQX impaired ASST performance in female and male pubertal mice and this effect was not specific to any of the ASST phases. With PEAQX treatment, pubertal mice actually showed an ASST performance very similar to those of adult mice (cf. Figure 1). In addition, there was a main effect of sex in this experiment, i.e., female mice performed better than male mice. This is in line with our first experiment and previous studies of our group with adult mice, in which often rather improved ASST performance in female mice was found [49,50]. Of note, the sex difference in this experiment with only pubertal mice reached statistical significance, whereas in our first experiment there was no sex difference in pubertal mice (*p* = 0.15) and a trend toward a sex difference in adult mice (*p* = 0.08). Furthermore, the bias for perseverative errors that we observed in the first experiment was not present in this third experiment. Thus, the PEAQX treatment had similar effects on both error types in both sexes. Future studies should address this inconsistency in sex differences and error bias.

To the best of our knowledge, this is the first time that the effects of a GluN2A-specific antagonist were tested in the ASST. In several other cognitive tests in adult mice, PEAQX (also named AAM-077) had minor or decreasing effects on learning performance and reversal learning was impaired [51,52]. Therefore, we would expect similar effects of PEAQX in adult mice than in the pubertal mice of the present study. In addition, it is of interest whether GluN2A-specific agonists are able to rescue impaired cognitive flexibility. To our knowledge, such an agonist has not been found so far. However, in rats, the unspecific partial NMDA receptor agonist D-cycloserine rescued pharmacologically induced deficits in the attentional set shifting task [11] and improved reversal learning [53].

## 5. Conclusions

The present study supports the previous observation in humans that cognitive flexibility is higher in puberty than in adulthood. Furthermore, our data show that cognitive flexibility is related to the expression of the NMDA receptor subunits GluN2A, GluN2B, and GluN2C in the orbitofrontal cortex and that subunit-specific pharmacological interventions (here, GluN2A) affect cognitive flexibility. Future studies should investigate whether subunit-specific agonists are able to rescue deficient cognitive flexibility and whether they have the potential to be used in human diseases with deficits in cognitive flexibility.

## Figures and Tables

**Figure 1 cells-12-01212-f001:**
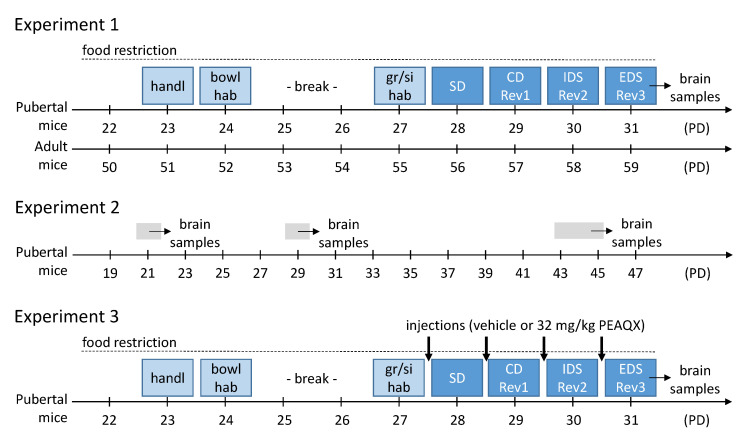
Experimental procedures of the three experiments. For details see main text. Abbreviations: bowl hab, habituation to bowl and setup; CD, compound discrimination; EDS, extra-dimensional shift; IDS, intra-dimensional shift; gr/si hab, group and single habituation; handl, handling; PD, postnatal day; Rev, reversal; SD, simple discrimination.

**Figure 2 cells-12-01212-f002:**
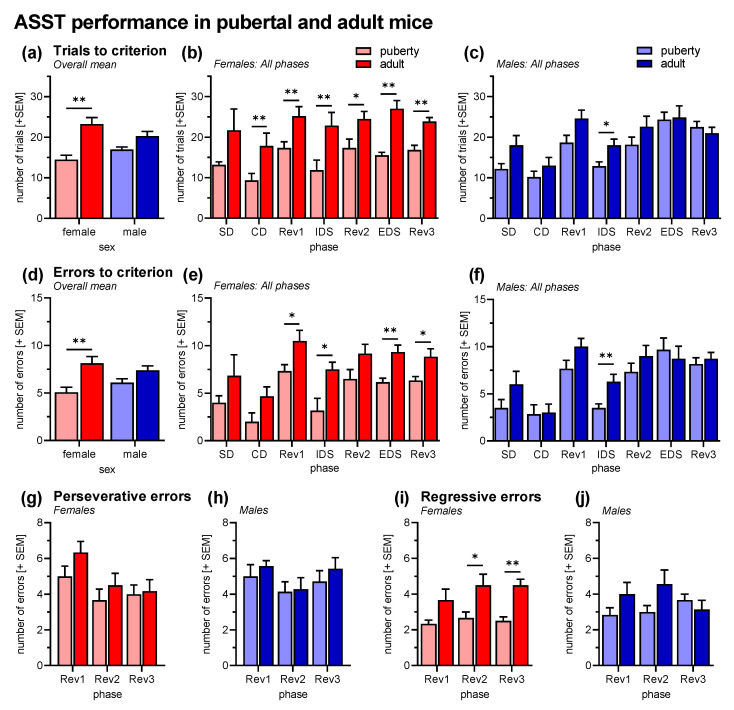
ASST performance in pubertal (*n* = 12) and adult mice (*n* = 13). (**a**) Overall trials to criterion were decreased in pubertal female mice but not in males. (**b**) This beneficial effect of puberty was observed in all ASST phases except SD in female mice. (**c**) In male mice, performance was only significantly improved in IDS. (**d**) This was mainly confirmed by the analysis of the errors to criterion; however, age did not interact with sex this time. (**e**) Nevertheless, in pubertal female mice, the errors were reduced in the Rev1, IDS, EDS, and Rev3 phases. (**f**) In pubertal male mice, improvements were only observed in IDS. (**g**,**h**) An error type analysis revealed that pubertal and adult mice did not differ in the number of perseverative errors. (**i**,**j**) However, regressive errors were reduced in pubertal female mice but not in male mice. ** *p* < 0.01, * *p* < 0.05, post hoc comparisons as indicated, after significant ANOVA effects.

**Figure 3 cells-12-01212-f003:**
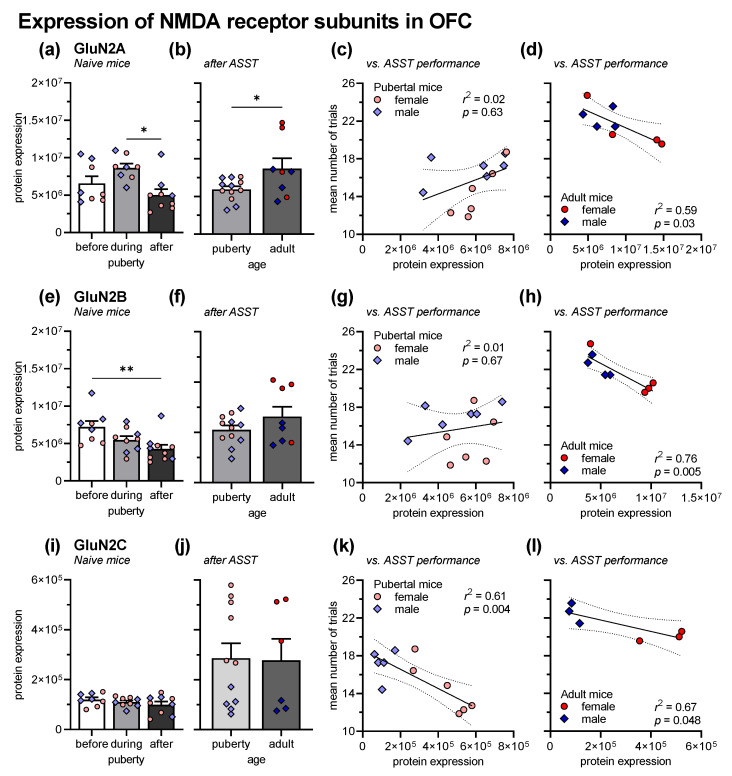
Expression of GluN2 in the OFC, in naive mice before (*n* = 8), during (*n* = 8) and after (*n* = 9) puberty and in pubertal (*n* = 11–13) and adult mice (*n* = 6–8) after ASST experience. The values for the individual mice are indicated by symbols (females = red circles; males = blue diamonds). (**a**) In naive mice, GluN2A expression was higher in pubertal mice than in adult mice. (**b**) This was opposite in ASST-experienced mice, i.e., GluN2A expression was higher in adult mice. Compared with naive mice, ASST experience decreased GluN2A expression in pubertal mice and increased it in adult mice. (**c**) In pubertal mice, GluN2A expression was not associated with ASST performance. (**d**) In adult mice, higher GluN2A expression was associated with lower trials to criterion, i.e., better ASST performance. (**e**) GluN2B expression decreased with age in naive mice, with lower expression in adult mice than before puberty. (**f**) In ASST-experienced mice, age had no effects on GluN2B expression. However, compared with naive mice, GluN2B expression was increased in adult mice. (**g**) In pubertal mice, GluN2B expression was not associated with ASST performance. (**h**) In adult mice, GluN2B expression positively correlated with ASST performance. (**i**) In naive mice, GluN2C expression was not affected by puberty. (**j**) The same was observed after ASST. However, compared with naive mice, GluN2C expression was generally increased. (**k**) GluN2C expression was correlated with better performance in pubertal mice. (**l**) In adult mice, higher GluN2C expression also led to less trials to criterion, i.e., better performance. ** *p* < 0.01, * *p* < 0.05, comparisons as indicated.

**Figure 4 cells-12-01212-f004:**
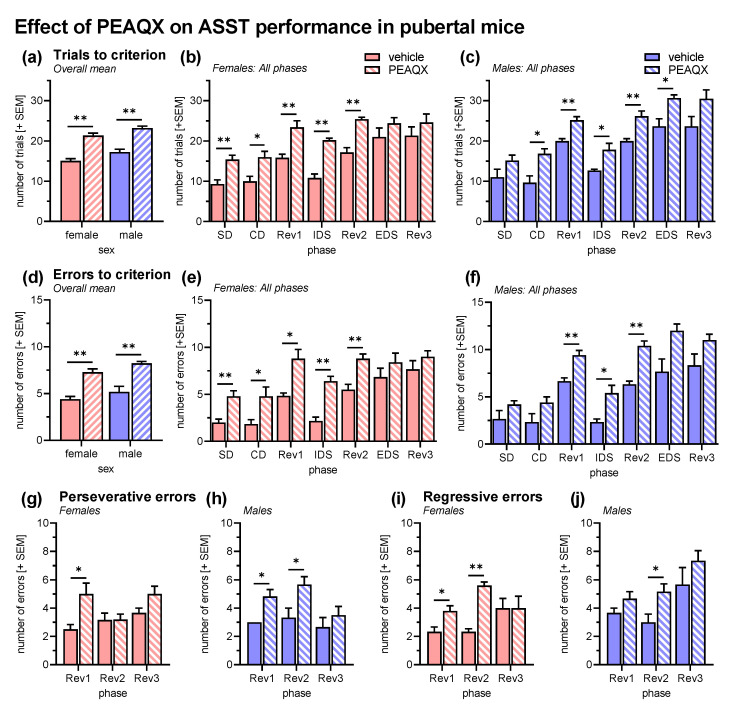
Effects of the GluN2A-specific antagonist PEAQX on the ASST performance in pubertal mice (vehicle: *n* = 9, PEAQX: *n* = 11). (**a**) Overall trials to criterion were increased after PEAQX treatment in pubertal female mice and male mice. (**b**) The PEAQX effect was observed in all ASST phases except EDS and Rev3 in female mice. (**c**) In male mice, performance was significantly decreased in Rev1, IDS, and Rev2. (**d**) These PEAQX effects were confirmed by the analysis of the errors to criterion, i.e., errors were increased in female and male mice. (**e**) In pubertal female mice, the errors were increased in all ASST phases except EDS and Rev3. (**f**) In pubertal male mice, increased errors were observed in REV1, IDS, and Rev2. (**g**,**h**) The error type analysis revealed that perseverative errors were increased by PEAQX, in female mice only in Rev1, in male mice in Rev1 and Rev2. (**i**,**j**) Similar effects were observed on regressive errors. They were increased in Rev1 and Rev2 of pubertal female mice and in Rev2 in male mice. ** *p* < 0.01, * *p* < 0.05, post hoc comparisons as indicated, after significant ANOVA effects.

## Data Availability

Data are contained within the article.

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
