# Peer review of "Cognitive Flexibility in Mice: Effects of Puberty and Role of NMDA Receptor Subunits"

_cells, 2023, doi:10.3390/cells12091212_

Round 1

Reviewer 1 Report

The paper “Cognitive flexibility in mice: effects of adolescence and role of NMDA receptor subunits “ written by Seifried  et al., investigates cognitive flexibility in mice dependent on age (adolescence vs adulthood) and sex (females and males). Moreover, cognitive flexibility was correlated with NMDA receptor subunits. This topic is interesting and might provide additional information related to the development of cognitive flexibility and its molecular background. However, there are some concerns related to the paper.

1.     Abstract: the sentence “ the GlN2A subunit …..was more highly expressed in adolescence” is not clear. The authors mean compare to adulthood and only in naïve mice. Do they analyze naïve adult mice?

2.     Abstract: the sentence .. to some extent, in adolescent mice,…..the expression of….correlated positively with performance. Only GluN2C is correlated in adolescence, maybe it should be pointed out.

3.     Materials and Methods. Some kind of schema for experiments should be included. There is not clear when exactly the PEAXQ was given ( 20 min before experiments, but which part). What were the beginning and final ages of mice with ASST experiments? Was the age correlated with the ages of naïve mice?

4.     Naïve mice PD 43-45 are suggested to be after adolescence. Are they adults? The ages PD21, PD29 and PD 43-45 are better fit for early adolescence, adolescence and late adolescence.  

5.     Based on the results it seems that changes in GluN2A receptors in adolescence are not crucial for better performance in adolescent female mice. Are there any possible explanations at molecular levels.

Author Response

REVIEWER 1

  1. Abstract: the sentence “ the GlN2A subunit …..was more highly expressed in adolescence” is not clear. The authors mean compare to adulthood and only in naïve mice. Do they analyze naïve adult mice?

ANSWER: We rephrased this sentence and use now the statement that “the GluN2A subunit of the NMDA receptor was more expressed during puberty than after puberty”. Based on comment 4 of this reviewer, we now decided to use the term puberty since we actually used pubertal mice for our behavioral experiments, and measured subunits levels before, during and after puberty and not before, during and after adolescence.

Unfortunately, we did not analyze the brains of adult mice in this study (8 weeks).

  1. Abstract: the sentence .. to some extent, in adolescent mice,…..the expression of….correlated positively with performance. Only GluN2C is correlated in adolescence, maybe it should be pointed out.

ANSWER: We agree and have reworded this sentence (lines 17-19) to read, “In adult mice, the expression of GluN2A, GluN2B, and GluN2C in the orbitofrontal cortex correlated positively with performance in the attentional set shifting task, whereas in pubertal mice this was the case for GluN2C.”

  1. Materials and Methods. Some kind of schema for experiments should be included. There is not clear when exactly the PEAXQ was given (20 min before experiments, but which part). What were the beginning and final ages of mice with ASST experiments? Was the age correlated with the ages of naïve mice?

ANSWER: We now included a new figure with a schematic representation of the different experiments (Figure 1). Drug administration was performed on each day of the ASST, 20 min before the start of the experimental session. This information is now also given in section 2.4.3 (line 239-240) and indicated in Figure 1.

The age of the pubertal mice used for ASST and for brain collection (group “during puberty”) was in the same range (see also Figure 1).

  1. Naïve mice PD 43-45 are suggested to be after adolescence. Are they adults? The ages PD21, PD29 and PD 43-45 are better fit for early adolescence, adolescence and late adolescence.

ANSWER: We agree with reviewer. Actually, the term puberty fits better, since PD21 is before, PD29 during, and PD43-45 after puberty. Therefore, we changed the wording in our manuscript and used now puberty/pubertal instead of adolescence/adolescent.

  1. Based on the results it seems that changes in GluN2A receptors in adolescence are not crucial for better performance in adolescent female mice. Are there any possible explanations at molecular levels.

ANSWER: We do not understand this comment. In experiment 3, administration of PEAQX, a GluN2A-specific antagonist, significantly reduced ASST performance in both, female and male pubertal mice. This clearly indicates that GluN2A is crucial for both sexes. By blocking them, the ASST performance of both sexes drops to the level of adults. 

Reviewer 2 Report

The authors used the ASST task to demonstrate that cognitive flexibility is higher in adolescent mice than in adult mice, effect that was more pronounced in female mice. Expression of GluN2A was higher during the adolescence than after adolescence, but after performance in the ASST the expression of GluN2A protein was lower in adolescent mice compared with adult mice. Generally, the manuscript is well written, but the gene expression results are difficult to interpret based on the study design.

Major comments:

The choice of the groups for the study of NMDA receptor subunits expression makes the description of the results inaccurate and the interpretation of the results difficult. The study in naïve mice does not include adult mice, the “after adolescence” groups consists of 6 weeks old mice that can not be considered as adults. It is therefore not clear on which groups of mice the comparison described on lines 292-293 was performed: “… while in adult mice, GluN2A expression was higher in ASST-experienced mice”.

A comparison between naïve adolescent mice and the ASST experienced adolescent mice is not possible due to the study design. It is not clear whether the experience of ASST or the general handling occurring during ASST is responsible for the decreased GluN2A expression in adolescent mice. A more appropriate control group would consist of mice handled the same way (food deprivation, ASST without reward; and reward presented un-paired with ASST, in the home cage after ASST for example) but without the cognitive flexibility task.  

The given PND for adolescence are not consistent, please verify and correct. On line 227, PND21 is defined as “before adolescence”, PND29 as “during adolescence” and PND43-45 as “after adolescence”, whereas on line 399 PND 35-40 is defined as onset of puberty and PND40-50 as adolescence. Is PND29 where the GluN2A expression was increased pre-puberty or adolescence? The discussed studies (References 21, 23, 24) that show contradictory results in cognitive flexibility to those described by the authors apparently used mice that were one week older. Please discuss a bit clearer and give the PND for comparison.

The ASST procedure should be described better:

1.       The food deprivation is poorly described. Adults mice were food deprived to 90-95% of their body weight, adolescent mice underwent a very moderate food restriction. How was this exactly done? Was the food deprivation performed for a certain amount of time, 24h or intermittent? Was the same protocol performed in all mice or a different protocol for each mouse based on body weight? What was the difference between food deprivation in adults and in adolescents? Why did the food deprivation protocol differ between adults and adolescents? How does this affect the behavioral results?

2.       Were the mice food deprived for the entire duration of the ASST (8 days) or only on day 1?

3.       Are the 6 different odors similarly pleasant/neutral for the mice?

4.       Can the mice smell the chocolate rice under the filling media?

5.       A schematic representation of the protocol might be useful.

Why was the GluN2A antagonist PEAQX administered i.p. (does it cross the blood-brain-barrier?) and not directly into the OFC where the expression of GluN2A was higher in adolescent mice? A general blockade of GluN2A subunits can be expected after i.p. administration and general effects on the brain. How is the expression of GluN2A subunits in other brain areas between adolescence and adulthood?

Minor comments:

Please explain what are the perseverative errors and the regressive errors.

Given that adolescent mice perform better in the task and have higher GluN2A expression in the OFC and antagonism of GluN2A impairs performance in adolescent mice, it is unclear why after performance in the task adolescent mice show reduced expression of GluN2A compared with adult mice that performed worse in the task. This does not seem logical and should be explained.

Please discuss why adolescent females perform better in the task compared to adolescent males, but in adults no such differences can be observed.

Author Response

REVIEWER 2

(…)

A comparison between naïve adolescent mice and the ASST experienced adolescent mice is not possible due to the study design. It is not clear whether the experience of ASST or the general handling occurring during ASST is responsible for the decreased GluN2A expression in adolescent mice. A more appropriate control group would consist of mice handled the same way (food deprivation, ASST without reward; and reward presented un-paired with ASST, in the home cage after ASST for example) but without the cognitive flexibility task. 

ANSWER: We are agree with the reviewer. These comparisons are not useful. We therefore removed the comparisons in section 3.2 and changed the respective part of the discussion (lines 460-472).

The given PND for adolescence are not consistent, please verify and correct. On line 227, PND21 is defined as “before adolescence”, PND29 as “during adolescence” and PND43-45 as “after adolescence”, whereas on line 399 PND 35-40 is defined as onset of puberty and PND40-50 as adolescence. Is PND29 where the GluN2A expression was increased pre-puberty or adolescence? The discussed studies (References 21, 23, 24) that show contradictory results in cognitive flexibility to those described by the authors apparently used mice that were one week older. Please discuss a bit clearer and give the PND for comparison.

ANSWER: Since we performed our behavioral tests in the phase of puberty (PD28-31), we now use pubertal and puberty throughout the manuscript. Then, “before, during and after puberty” in the naive mice is correct. Importantly, we would like to point out to this reviewer that PD35-40 and PD40-50 are referred to in our discussion as the “offset” (not the onset!) of puberty.

The ASST procedure should be described better:

  1. The food deprivation is poorly described. Adults mice were food deprived to 90-95% of their body weight, adolescent mice underwent a very moderate food restriction. How was this exactly done? Was the food deprivation performed for a certain amount of time, 24h or intermittent? Was the same protocol performed in all mice or a different protocol for each mouse based on body weight? What was the difference between food deprivation in adults and in adolescents? Why did the food deprivation protocol differ between adults and adolescents? How does this affect the behavioral results?

    Answer: The food restriction procedure is now better described in section 2.3.2 (lines 118-123). The food restriction was until the end of the experiment. The amount of food was based on the individual body weight of the mice, and the food was given every day after the behavioral experiment. The difference between adults and pubertal mice was that the adult mice slightly decreased their body weight (minus 5-10%) whereas pubertal mice maintained their body weight.
  2. Were the mice food deprived for the entire duration of the ASST (8 days) or only on day 1?

Answer: Mice were food-deprived for the entire ASST. This information is now given in section 2.3.2 (line 119) and in Figure 1.

  1. Are the 6 different odors similarly pleasant/neutral for the mice?

Answer: During the last 5 years, we observed that all six odors (the media as well) are similarly quickly learned if used during the simple discrimination phase. Further, we never observed obvious aversive or appetitive responses of the mice. Therefore, we are confident that the odors and media are similarly pleasant/neutral for the mice.

  1. Can the mice smell the chocolate rice under the filling media?

      Answer: We added now the information in section 2.3.5 (lines 143-145) that unbaited bowl was sprinkled with Choco Rice powder to prevent the mice from using the smell to retrieve the reward. This is usual in mice ASST experiments. Further, the higher error rate in the reversal phases demonstrates that the mice do not simply use the smell of the reward.

  1. A schematic representation of the protocol might be useful.

Answer: We added figure 1 showing the protocols of the experiments.

Why was the GluN2A antagonist PEAQX administered i.p. (does it cross the blood-brain-barrier?) and not directly into the OFC where the expression of GluN2A was higher in adolescent mice? A general blockade of GluN2A subunits can be expected after i.p. administration and general effects on the brain. How is the expression of GluN2A subunits in other brain areas between adolescence and adulthood?

Answer: PEAQX crosses the blood-brain barrier, as demonstrated by several experiments showing blockade of epileptoform neuronal activity and seizures triggered by it. These studies are cited now in section 2.2 (lines 92 & 95). Repeated local infusions into the OFC in 4 weeks old mice via chronically implanted cannulas are very difficult and are not established in our laboratory (and probably elsewhere). Most probably, local down-regulation by viral constructs would be a more promising approach.

We do not know whether the differences between before, during, and after puberty that we observed in the OFC are also present in other brain regions involved in ASST. We assume that the differences are similar but this needs to be tested in future analyses.

Minor comments:

Please explain what are the perseverative errors and the regressive errors.

ANSWER: We are sorry about this negligence. We have added a section “2.6 Statistical analyses” (lines 248-259) explaining the classification of error types, as well as the general statistical analyses.

Given that adolescent mice perform better in the task and have higher GluN2A expression in the OFC and antagonism of GluN2A impairs performance in adolescent mice, it is unclear why after performance in the task adolescent mice show reduced expression of GluN2A compared with adult mice that performed worse in the task. This does not seem logical and should be explained.

ANSWER: Based on the first comments of this reviewer, we removed the comparison of expression levels in naive and ASST-experienced mice since we follow the reviewer’s opinion that a comparison is not possible due to the study design. Actually, the difference between the two groups is not only the cognitive challenge by the ASST but also handling, food restriction, digging, eating reward, etc. - as discussed in lines 463-471.

Please discuss why adolescent females perform better in the task compared to adolescent males, but in adults no such differences can be observed.

ANSWER: In our third experiment, we observed this sex difference in pubertal mice. However, in our first experiment, we actually could not find such a sex difference in pubertal mice while there was a trend for a sex difference in adult mice. This is now mentioned and shortly discussed in the revised version of the manuscript (line 496-499).

Reviewer 3 Report

The manuscript title “Cognitive flexibility in mice: effects of adolescence and role of NMDA receptor subunits.” The data indicate that GluN2A expression in the OFC is associated with higher cognitive flexibility during adolescence. This study has a precise aim and reasonable rationale. However, several concerns need attention.

 1.      During adolescence, there was only a trend for reducing perseverative errors, while regressive errors decreased significantly in female mice. PEAQX increased both perseverative and regressive errors. What is the meaning of the results? Authors should discuss it.

2.      Ip injection of PEAQX may affect many regions of the central nervous system. What is the effect of PEAQX on OFC or PFC? Authors should discuss it.

3.      Page 5, Line 211: “NMDAR2C” change to "anti-NMDAR2C"?

4.      Page 5, Line 213: add “IgG”?

5.      Page 5, Line 247: “inter with” change to “interact with”?

Author Response

REVIEWER 3

Comments and Suggestions for Authors

(…)

  1. During adolescence, there was only a trend for reducing perseverative errors, while regressive errors decreased significantly in female mice. PEAQX increased both perseverative and regressive errors. What is the meaning of the results? Authors should discuss it.

Answer: We agree with the reviewer. Actually, the error types were not discussed at all so far. We now added sections in lines 414-425 and 499-502.

  1. Ip injection of PEAQX may affect many regions of the central nervous system. What is the effect of PEAQX on OFC or PFC? Authors should discuss it.
  2. Page 5, Line 211: “NMDAR2C” change to "anti-NMDAR2C"?

      Answer: This was corrected.

  1. Page 5, Line 213: add “IgG”?

      Answer: “antibodies” was added.

  1. Page 5, Line 247: “inter with” change to “interact with”?

      Answer: This was corrected.

Round 2

Reviewer 1 Report

The manuscript has been improved. I would suggest adding the reference {37, Scheider M} in 2.4.2 part of Materials and Methods, after the sentence "(before,  during and after puberty, Figure 2.)"